# Hydrogen Bonding and Solvation of a Proline-Based Peptide Model in Salt Solutions

**DOI:** 10.3390/life11080824

**Published:** 2021-08-12

**Authors:** Sara Catalini, Barbara Rossi, Mariagrazia Tortora, Paolo Foggi, Alessandro Gessini, Claudio Masciovecchio, Fabio Bruni

**Affiliations:** 1European Laboratory for Non-Linear Spectroscopy, LENS, Via Nello Carrara, 1, 50019 Sesto Fiorentino, Italy; catalini@lens.unifi.it (S.C.); paolo.foggi@unipg.it (P.F.); 2Elettra-Sincrotrone Trieste, S.S. 114 km 163.5, Basovizza, 34149 Trieste, Italy; mariagrazia.tortora@elettra.eu (M.T.); alessandro.gessini@elettra.eu (A.G.); claudio.masciovecchio@elettra.eu (C.M.); 3Area Science Park, Padriciano, 99, 34149 Trieste, Italy; 4Dipartimento di Chimica, Biologia e Biotecnologie, Università di Perugia, Via Elce di Sotto, 8, 06123 Perugia, Italy; 5Dipartimento di Scienze, Università degli Studi Roma Tre, Via della Vasca Navale, 84, 00146 Roma, Italy; fabio.bruni@uniroma3.it

**Keywords:** peptide, hydrogen bonding, proline, UV resonance Raman

## Abstract

The hydrogen bonding of water and water/salt mixtures around the proline-based tripeptide model glycyl-l-prolyl-glycinamide·HCl (GPG-NH_2_) is investigated here by multi-wavelength UV resonance Raman spectroscopy (UVRR) to clarify the role of ion–peptide interactions in affecting the conformational stability of this peptide. The unique sensitivity and selectivity of the UVRR technique allow us to efficiently probe the hydrogen bond interaction between water molecules and proline residues in different solvation conditions, along with its influence on *trans* to *cis* isomerism in the hydrated tripeptide. The spectroscopic data suggest a relevant role played by the cations in altering the solvation shell at the carbonyl site of proline_._, while the fluoride and chloride anions were found to promote the establishment of the strongest interactions on the C=O site of proline. This latter effect is reflected in the greater stabilization of the *trans* conformers of the tripeptide in the presence of these specific ions. The molecular view provided by UVRR experiments was complemented by the results of circular dichroism (CD) measurements that show a strong structural stabilizing effect on the β-turn motif of GPG-NH_2_ observed in the presence of KF as a co-solute.

## 1. Introduction

The question of how a given polypeptide sequence folds into the three-dimensional biologically active structure of the protein is still an intriguing problem in biophysics. The stability of peptides and proteins in a solution results from a delicate balance between different interactions, e.g., hydrophobic forces, electrostatic interactions and hydrogen bonds (HB). Various factors can affect this equilibrium, such as ions in the solution. The non-covalent interaction of ions and proteins in an aqueous solution plays a key role in the thermodynamics and kinetics of protein folding [1]. Salts can affect the proteins’ conformation through two major factors: (i) direct interaction with the protein surface and the polypeptide side chain groups and (ii) perturbation of the hydration shell around the biomolecule. It has been shown that the salts can act as stabilizing or destabilizing agents of polypeptide and protein structures in a water solution, exerting both non-polar (Hofmeister) and electrostatic effects [2,3,4,5]. As suggested by experimental and theoretical work, certain inorganic salts favor the formation of partly structured states in protein folding, acting as potent stabilizers of the native state [6]. However, how the ions interact with peptides and proteins to induce Hofmeister effects also remains to be fully clarified [7]. Some studies performed on model amide systems [3,8,9] gave evidence that the interactions between anions and the peptide bonds are modulated by adjacent non-polar groups [10,11]. Other works suggested the dominant role played by the protein surface charge in ion–protein interactions that could explain the reversed Hofmeister sequence observed for the capacity of some anions to stabilize protein structures in a solution [12]. Considering the complexity of the interactions that may occur between proteins and ions, a deeper understanding of the solvation and hydrogen bonding properties of model peptides in salt solutions would be useful for determining the forces that drive the folding path of proteins.

In polypeptide sequences, the amino acid proline (Pro) is considered to have a unique role in determining protein structure and folding. Proline residues are frequently found at bends in the polypeptide chain, and they play a structurally specific role in β-turn [13,14] thanks to their reduced flexibility, which is ideally suited for reversing the direction of the polypeptide chain. The presence of a side-chain pyrrolidine ring severely restricts the rotation around the C–N bond, while the absence of a hydrogen atom on the imide nitrogen prevents its participation in hydrogen bonding [15]. Another unique property of Pro residues is that the energy barrier of rotation of the peptide linkage, X-Pro, of proline with the preceding X-amino acid is significantly lower compared to the regular amide bonds [16]. This implies that the *trans–cis* isomerization of the X-Pro bond occurs more easily than the usual peptide bonds. The conformational flexibility of the imide bond of proline has been observed to be an important factor in protein dynamics and folding [17,18]. For instance, it has been found that proline-rich regions of the protein immunoglobulin-G are strongly involved in antigen binding by mediating a structural transition in the hinge region of the protein [19]. *Trans–cis* isomerization about the proline bond is believed to be critical for protein stability [20] because the interconversion between the two isomeric forms is one of the rate-determining steps in protein folding [17,21]. These studies suggested that proline residues can act as a configuration controller in proline-containing proteins, playing a key role in specific biochemical processes [22]. The *trans–cis* isomerization of the X-Pro bond is of interest for its implication in protein denaturation processes [17,23,24,25], since the incorrect X-Pro isomers in the unfolded forms of proteins has been observed. Some investigations showed that proline isomerization can induce long-range perturbations in the secondary structure of polypeptide chains, thus suggesting the possible function of proline as a molecular switch in proteins [26]. Finally, the *trans–cis* isomerization of proline has been found to be involved in the gating mechanism of ion channels and in the biomolecular recognition processes of some proteins [27,28].

Multi-wavelength UV resonance Raman (UVRR) spectroscopy is a powerful method for investigating the hydrogen bonding in peptides and proteins, and for directly identifying switching between the *trans* to *cis* isomers that is not easily detected by other methodologies [29,30,31,32,33]. Accurate tuning of the UV excitation energy allows one to enhance, in the UVRR spectra, the Raman signals associated with the X-Pro bond vibrations, thanks to the X-Pro amide absorption maximum that is red-shifted by about 10 nm in comparison with that of other backbone amides [30,31,32,33]. This provides a clear identification of the distinct Raman markers of both the conformation and the HB state of the amide sites present in the polypeptide chain. The UVRR technique can provide a direct spectroscopic determination of the *trans–cis* equilibrium of the X-Pro bond in peptides and proteins [31,32], which is useful for understanding the role of Pro isomerization in driving peptides’ and proteins’ conformation.

In the present study, the influence of different salts on hydrogen bonding and on the structure of a simple proline-based tripeptide model was investigated to address the following questions: (i) Do the cations and anions exhibit the preferred HB interaction with the cis or trans conformers of peptide? (ii) How do the ion interactions affect the trans–cis equilibrium of the proline-based peptide in different salt solutions? (iii) How can this be related to the thermal structural stability of the peptide? The tripeptide glycyl-l-prolyl-glycinamide·HCl (GPG-NH_2_) has been chosen for this study as a very simplified model system which contains the glycine–proline–glycine sequence that is a common sequence in the structural motifs of β-turns [34,35]. The structure of GPG-NH_2_ in an aqueous solution has been previously assessed using NMR spectroscopy and molecular dynamic (MD) simulations [34]. Those experiments estimated an 85:15 mixture of *trans:cis* conformations (Figure 1) about the Gly1–Pro2 peptide bond. The hydration of GPG-NH_2_ has been investigated at atomic-scale level [34,35], finding that when the tripeptide is in *trans* form, a water-mediated HB between the oxygen of Gly and the NH_2_ moiety of the other Gly residues seems to guide the folding nucleation step [34]. However, a more recent study also highlighted a direct non-water-mediated bond between these two sites [35]. Both studies suggested that the proline ring has a steric influence on the hydration shell of Gly1–Pro peptide bond oxygen, regardless of the isomeric bond’s conformation [34,35]. The sensitivity and selectivity of the synchrotron-based UVRR technique allowed us to probe the structure and the strength of HB interactions between the water molecules and specific sites of GPG-NH_2_, e.g., the C=O group of the Pro residue and of other amide sites, under different experimental conditions. The influence of the ions in solution on the *trans–cis* isomerization equilibrium of Pro residues was also explored. The overview offered by the UVRR data was complemented by circular dichroism (CD) measurements that provided insights on the conformational variations of the GPG-NH_2_ tripeptide induced by the increase in temperature in the presence and absence of salts.

## 2. Materials and Methods

The tripeptide glycyl-l-prolyl-glycinamide HCl (GPG-NH_2_, MW = 264.71 g/mol, >99% purity) was purchased from Bachem and used without further purification. No relevant contamination by water could be measured in the lyophilized peptide powder, as deducted by the absence in the Raman spectra of any signal attributable to the intense OH stretching band of water. The salts considered in this study (NaCl, NaBr, KCl, KBr and KF) were acquired from DBA Italia. Aqueous solutions of GPG-NH_2_ were prepared by dissolving the dry peptide in doubly distilled deionized water in order to obtain the desired concentrations of peptide in water, ranging from about 294 to 48 mg/mL (corresponding to a molar ratio of 1:58 GPG-NH_2_:H_2_O and 1:309 GPG-NH_2_: H_2_O, respectively). The deuterated GPG-NH_2_ solution was obtained by dissolving the tripeptide in deuterium oxide (99.9 atom % D, Sigma-Aldrich Chemie GmbH, Schnelldorf, Germany. The ternary GPG-NH_2_/H_2_O/salt solutions were prepared by dissolving a suitable amount of tripeptide and salt in the doubly distilled deionized water to achieve the desired concentrations (corresponding to a molar ratio of 1:309:14 GPG-NH_2_: H_2_O: salt). All the solutions were measured as a function of temperature, ranging from 298 to 373 K.

Raman spectra were collected using the Synchrotron radiation UV resonance Raman (SR-UVRR) set-up available at the BL10.2-IUVS beamline of Elettra Sincrotrone Trieste, Trieste (Italy) [29]. The exciting wavelength was set at 226 nm by regulating the undulator gap and using a Czerny-Turner monochromator (Acton SP2750, Princeton Instruments, Acton, MA, USA) equipped with a holographic grating with 3600 groves/mm for monochromatizing the incoming synchrotron radiation. The excitation radiation at 266 nm was provided by a CryLas FQSS 266-Q2, Diode Pumped Passively Q196 Switched Solid State Laser. UVRR spectra were recorded in back-scattered geometry by using a single pass of a Czerny–Turner spectrometer (Trivista 557, Princeton Instruments, 750 mm of focal length) provided with a holographic grating at 1800 g/mm. The calibration of the spectrometer was standardized using cyclohexane (spectroscopic grade, Sigma Aldrich). Any possible photo-damage effects due to prolonged exposure of the sample to UV radiation were avoided by continuously spinning the sample cell during the measurements.

Circular dichroism (CD) spectra were recorded using a Jasco J-810 polarimeter equipped with a plug-and-play single-cell Peltier with a stirrer for temperature control. Both the binary and ternary solutions were measured in a quartz cell with a 1 mm path length (concentration of GPG-NH_2_ of 0.1 mg/mL). Each CD spectrum was collected in the range from 185 to 280 nm in increments of 0.5 nm, with a scan rate of 20 nm/min and a bandpass of 1 nm. All the spectra were averaged over 8 scans. We performed the measurements under a constant nitrogen flow to purge the ozone generated by the light source of the instrument. The CD spectra of the GPG-NH_2_/H_2_O and GpG-NH_2_/H_2_O/KF solutions were collected at different temperatures ranging from 298 to 363 K with ΔT = 5 K. For each set of temperature-dependent measurements, the CD spectrum of the buffer alone (water and water/salt) was subtracted from the spectra of the corresponding samples. The temperature evolution of the CD spectra was monitored by evaluating the ellipticity θ_λ_ at 200 nm in the explored temperature range.

## 3. Results

The multi-wavelength UVRR spectra of peptides collected using different excitation wavelengths are rich in information thanks to the quite different UV absorption bands of the various chromophores of the polypeptide chains [29,36,37,38]. Figure 2 displays the UVRR spectra obtained with 266 and 226 nm as the excitation wavelengths of the lyophilized powder of GPG-NH_2_ and of the aqueous solutions of the tripeptide.

The main Raman signals that could be identified in the spectrum of the lyophilized powder of GPG-NH_2_ were also visible in the vibrational profiles of the peptide dissolved in light and heavy water collected at an excitation wavelength of 266 nm (three traces at the bottom of Figure 2). The same Raman signals appeared to be broader and slightly blue-shifted in frequency in the spectrum of the aqueous solution of the tripeptide with respect to the crystalline form of GPG-NH_2_. This is ascribed to the formation of HB between the surrounding water molecules and specific sites of the peptide [37]. In the wavenumber region between 1200 and 1800 cm^−1^, it is possible to identify the AI band (1600–1700 cm^−1^), which is mostly attributed to the C=O stretching vibration and the AII band (1500–1600 cm^−1^) that results from the combination of N–H bending and C–N stretching [37]. Both these Raman signals appear to be split into two spectral components spaced by ~30–40 cm^−1^ in the spectrum of lyophilized GPG-NH_2_ powder. This finding indicates that the amide bonds present in the tripeptide’s structure were differently involved in HB. The spectral range between 1200 and 1400 cm^−1^ in the 266 nm-excited spectra of lyophilized and hydrated GPG-NH_2_ (Figure 2, traces at the bottom) contained several Raman signals that could be assigned to the AIII bands. These vibrational modes arise from the combination of N–H bending and C–N stretching vibration. The comparison between the Raman spectra of the tripeptide dissolved in light and heavy water displayed the significant modification of the AIII band region, which was expected since the strong contribution of the N–H bending vibration to AIII band signals. Conversely, the frequency position of the AI band was not notably affected by the H/D isotopic exchange because of its strong C=O stretching character. The prominent signal observed at 1466–1476 cm^−1^ in the 266 nm-excited spectra of lyophilized and hydrated tripeptide in Figure 2 is attributable to the AII_p_ band of the X-Pro peptide bond [30,31,32,33]. The composition of this vibrational mode is similar to that of the AII vibration without the coupling between the C–N stretching and N–H bending displacement that is characteristic of amide signals [30]. The strong character of the localized C–N stretching vibration of the AII_p_ band was reflected by the distinct frequency position of the corresponding Raman signal, which fell at about 1470 cm^−1^ in the spectra of GPG-NH_2_. A comparison between the 266 nm-excited spectra of the tripeptide dissolved in H_2_O and D_2_O gave evidence that the Raman signal observed at 1476 cm^−1^ in the H_2_O solution of GPG-NH_2_ was not affected by the isotopic exchange. This is consistent with the assignment of this band to the AII_p_ band of the X-Pro peptide bond. Since GPG-NH_2_ dissolved in water contains both the *cis* and *trans* isomeric forms of the peptide [34,35], we associated the Raman signals observed at 1433 and 1476 cm^−1^ in the spectrum of the hydrated tripeptide with the vibrations of the proline AII_p_ band of the *cis* (c-AII_p_) and the *trans* (t-AII_p_) conformers, respectively [30,31]. Our attribution is consistent with the assignments reported in previous works for *cis* and *trans* conformers of polyproline and other dipeptides containing proline [30,31]. Moreover, the downshift of ≈40 cm^−1^ in the proline AII_p_ band of the *cis* form of GPG-NH_2_ compared to the *trans* one is in agreement with the observation that the amide II frequency of *cis* peptides is downshifted compared with that of *trans* peptides [39]. Interestingly, the Raman signals ascribable to the c-AII_p_ and t-AII_p_ bands are both red-shifted by about 10 cm^−1^ in the spectrum of the lyophilized powder of GPG-NH_2_ with respect to the GPG-NH_2_/H_2_O solution. This downshift can be related to the weakening of hydrogen bonding on the carbonyl group of the Pro residue [30,33] that occurred when GPG-NH_2_ was in its crystalline form in contrast to the interactions involved between the carbonyl group and the surrounding water molecules in the hydrated tripeptide. The behavior of the AII_p_ band in the tripeptide GPG-NH_2_ can be clearly distinguished from that of the other common amide Raman bands by exploiting the different absorption properties of the X-Pro bond. Indeed, the 226 nm-excited Raman spectrum of the GPG-NH_2_/H_2_O solution (trace at the top in Figure 1) showed the notable enhancement of the t-AII_p_ and c-AII_p_ signals compared with the UVRR spectra of the hydrated GPG-NH_2_ recorded with the 266 nm wavelength. Using an excitation wavelength at 226 nm, which is closer to the resonance conditions of the X-Pro bond, we observed an enhancement of about threefold in the AII_P_ band with respect to the Raman signals assigned to other amide bands [40]. This effect has been attributed to the bathochromic shift in the $\pi -\pi^{*}$ electronic transition of the X-Pro peptide linkage [30,31,40].

## 4. Discussion

### 4.1. Temperature-Dependence of Hydrogen Bonding on the Amide Bonds of GPG-NH_2_

Figure 3 displays the temperature evolution of the 266 nm-excited spectra of the GPG-NH_2_ dissolved in neat water. From inspection of the figure, it is evident that the rise in temperature induced a slight upshift of the AI band and a downshift of both the t-AII_P_ and the AIII bands. The insets (a–c) in Figure 3 report the temperature trends of the central wavenumber positions of the three Raman bands. Since the amide signals are very sensitive to interactions involving the peptide amide sites, the temperature behavior of the AI and AIII bands reflects the thermally induced changes in the HB between the C=O and C–N groups of GPG-NH_2_ and the water molecules in the hydration shell. The linear frequency shift observed for AI and AIII Raman signals (insets (a) and (c) in Figure 3) reflects the effect of thermal motion on the oscillator strength associated with the peptide link [41,42] that is modulated by the perturbation induced on the hydrogen bonding at the carbonyl and amide sites of GPG-NH_2_. The same effect has been observed previously in other peptide–water systems [37,38,42].

In particular, the blue-shift of the AI band and the red-shift of the AIII signal, upon the increase in temperature, suggest a progressive weakening of the HB between water molecules and the C=O and N–H groups of the tripeptide, due to the increase in thermal motion. The opposite temperature-trend found for the other two amide bands was expected, based on the different normal-mode composition of these bands [41,42].

The proline amide band AII_p_ is a very sensitive marker of the H-bonding state at the C=O site of the proline imide bond [30,31,33]. The frequency position of this Raman band can be a structural marker of X-Pro bonds for distinguishing different hydrogen bonds with solvents or other amino acid residues and the C=O group of proline [30,33]. The downshift of about 3 cm^−1^ observed for the Raman signal t-AII_p_ of hydrated GPG-NH_2_ can be ascribed to the progressive loss of inter- and intra-molecular hydrogen bonding on the X-Pro carbonyl site promoted by the increase in temperature.

### 4.2. Effect of Salts on the Hydrogen Bonding at the Carbonyl Site of Pro

UV excitation at 226 nm selectively enhanced the Raman bands of proline at ≈ 1470 cm^−1^, which dominated the spectrum of GPG-NH_2_, as visible in Figure 2. Although the GPG-NH_2_ contains only one X-Pro bond, the Raman band associated with this molecular moiety could be clearly detected against the large background of other peptide bonds in the UVRR spectra excited at 226 nm. Figure 4 displays the 226 nm-excited Raman spectra of the binary peptide/water and the ternary peptide/water/salt solutions of GPG-NH_2_ recorded at 298 K. In each ternary solution, there are 14 molecules of salt for one molecule of the peptide. A comparison among the UVRR spectra in Figure 4 suggests that different solvation environments around the GPG-NH_2_ peptide induced a change in the t-AII_p_ band’s frequency position. In particular, the presence of ions in the hydration shell of the peptide was marked by an upshift in the wavenumber position of the t-AII_p_ Raman signal. Solvation studies on peptide models of the X-Pro bond [30] have shown that the wavenumber position of AII_p_ increases linearly with the solvent acceptor number. The blue-shift of the t-AII_P_ band observed in Figure 4 for the ternary solutions suggests that the C=O site of proline in GPG-NH_2_ is involved in stronger HBs when the peptide is dissolved in water/salt mixtures compared to the pure water solvent. This is reflected by the notable change in the solvation properties of the tripeptide GPG-NH_2_ due to the addition of salts in the solution.

Table 1 reports the frequency positions of the t-AII_p_ band for the ternary solution of GPG-NH_2_ and salts, and the difference Δν estimated with respect to the position of the t-AII_p_ band in pure water. The upshift, Δν, of the t-AII_p_ band induced by the addition of salt measures the magnitude of hydrogen bond donation to the carbonyl group of the proline as a function of the different ion types [30,33].

The potassium salts (KCl and KBr) induced the formation of stronger carbonyl hydrogen bonding on the proline of GPG-NH_2_ in comparison with their sodium analogs (NaCl and NaBr), as detected by the more marked upshift of the t-AII_p_ band (see Table 1). This advices the notable role played by the cations on the hydrogen bonding and solvation properties of the tripeptide GPG-NH_2_. It is noteworthy that the more pronounced upshift of the proline amide band was induced by the addition of KCl and KF salts to a water solution of GPG-NH_2_. This finding seems consistent with the effect of the halide anions on the peptides’ structure, as already reported in the literature [43,44].

Figure 5 displays the temperature dependence of the wavenumber position of the t-AII_p_ band of hydrated GPG-NH_2_ in the absence and presence of salt co-solutes.

The thermally induced red-shift of the proline amide signal in GPG-NH_2_ gives an indication of an overall reduction in the strength of HBs involving the C=O site of the X-Pro bond upon the increase in temperature. Interestingly, the effect observed at room temperature by the addition of salts seemed to be maintained throughout the entire temperature range.

### 4.3. Influence of Salts on Trans/Cis Isomerization

The *trans–cis* equilibrium in the tripeptide GPG-NH_2_ can be efficiently probed through an analysis of the UVRR signals assigned to the *cis* and *trans* conformers of proline, i.e., the c-AII_p_ and t-AII_p_ bands, respectively. The intensities of these Raman signals reflect the population of the corresponding *cis* and *trans* conformers of GPG-NH_2_ at a certain temperature. The free energy difference, $\Delta G_{cis-trans}$, between the *cis* and *trans* conformations can be estimated by determining the temperature dependence of the *cis* and *trans* Raman bands’ intensity ratio [45,46]:(1)$$\ln\left(\frac{I_{c-\mathrm{AIIp}}}{I_{t-\mathrm{AIIp}}}\right)=\ln\left(\frac{\sigma_{\mathrm{cis}}}{\sigma_{\mathrm{trans}}}\right)-\frac{\Delta G_{cis-trans}}{\mathrm{RT}}$$
where $I_{c-\mathrm{AIIp}}$ and $I_{t-\mathrm{AIIp}}$ are the intensities of the *cis* and *trans* AII_p_ signals, $\sigma_{\mathrm{cis}}$ and $\sigma_{\mathrm{trans}}$ represent the AII_p_ Raman cross-sections of the *cis* and the *trans* conformers, R is the gas constant and T is the temperature. Figure 6a,b displays the plot of the log of the *cis* and *trans* AII_p_ intensity as a function of 1/T of GPG-NH_2_ hydrated in pure water and in water/salt mixtures. The linear fit of the data allowed us to estimate the free energy difference, $\Delta G_{cis-trans}$, between the *cis* and *trans* conformers of GPG-NH_2_ in the different solvation environments (Figure 6c). Our results showed a 2.5 ± 0.3 kJ/mol free energy difference of GPG-NH_2_ in the water/NaCl solution, a value of 1.8 ± 0.5 kJ/mol in the water/NaBr solution, a value of 4.2 ± 0.3 kJ/mol in the water/KF solution, a value of 3.9 ± 0.3 kJ/mol in the water/KCl solution and a value of 2.2 ± 0.4 kJ/mol in the water/KBr solution.

It is noteworthy that the tripeptide GPG-NH_2_ dissolved in the KF and KCl solutions appeared to have the highest energy difference between the *cis* and *trans* forms, while for the pure water and the other water/salt solutions, the energy difference between the two conformers was found to be the same, given the experimental error. The greater impact on the *cis–trans* energy difference exerted by the KF and KCl salts could be ascribed to the alterations of the solvation shell around the C=O site of proline observed when these salts were added to the hydrated GPG-NH_2._ Indeed, the stronger HB involving the proline residue could act by stabilizing the *trans* form of the peptide hydrated in KF and KCl solutions. This result seems to be consistent with the influence of Hofmeister ions on the thermodynamics of the *trans–cis* equilibrium in a proline-based peptide model previously described by other authors [2].

### 4.4. Effect of Salts on the Structure of the Tripeptide GPG-NH_2_

The analysis of CD spectra of GPG-NH_2_ in different solvent conditions provided insights into the conformational changes induced in the tripeptide by the presence of co-solvents. Figure 7a,b displays the temperature evolution of the CD profiles of GPG-NH_2_ dissolved in pure water and in a water/KF solution in the region of the $\pi -\pi^{*}$ and $n-\pi^{*}$ transitions of amidic groups.

At the lowest temperature value, the CD spectra of both the GPG-NH_2_ in water and the water/salt solution exhibited a negative ellipticity at about 200 nm and a broad positive band from 210 to 220 nm. These features, which were already observed in the CD spectra of small peptides containing proline, have been interpreted as the spectroscopic markers of β-turn structures [47]. As the temperature increased, the intensity of the minimum lessened for both samples of GPG-NH_2_ in neat water and in the aqueous salt solution, suggesting the occurrence of a progressive loss of the β-turn conformation of the tripeptide induced by the thermal motion. Interestingly, some differences appeared in the temperature evolution of the negative band at 200 nm, depending on the solvent conditions (see the comparison between the spectra in Figure 7a,b). It is possible to follow the relation between the temperature and the β-turn structure of GPG-NH_2_ by monitoring the ellipticity θ_λ_ at 200 nm, as seen in the insets of Figure 7a,b. These curves describe a characteristic melting pathway for the tripeptide that was notably affected by KF as a co-solute. The trends reported in the insets of Figure 7 show a two-state transition with a transition temperature of about 339 K for GPG-NH_2_ in water and 358 K for GPG-NH_2_ in the water/KF mixture. This result confirms the stabilizing effect on the structure of the tripeptide GPG-NH_2_ exerted by the salt KF, in agreement with the altered hydrogen bonding solvation shell at the proline C=O site detected by UVRR measurements for this salt.

## 5. Conclusions

The solvation properties and the hydrogen bonding of a proline-based peptide model were investigated here in the presence of salt co-solutes to address the role of ion–peptide interactions in affecting the conformational stability and the *trans–cis* equilibrium of this biomolecule. Multi-wavelength UVRR spectroscopy provides a unique sensitivity to the hydrogen bonding of proline residues in polypeptides, particularly in distinguishing the different HB networks around the peptide backbone site of X-Pro compared with the amidic sites that involve other amino acid residues. The analysis of specific Raman signals, that can be enhanced by tuning the excitation wavelength, allows one to probe the effect of ions on the *trans* to *cis* isomerization of the X-Pro bond in the tripeptide. Our Raman data suggest that the C=O site of proline in GPG-NH_2_ is involved in stronger hydrogen bonds in the hydration shell of the tripeptide when the salts are added than in the neat water case. Interestingly, we found that potassium salts (KCl and KBr) induced the formation of stronger hydrogen bonding at the carbonyl group of proline in comparison with their sodium analogs (NaCl and NaBr), suggesting the non-negligible role played by the cations on the solvation shell around the tripeptide. The relevant impact exerted by the fluoride and chloride anions of the potassium salts in altering the hydrogen bonding state at the C=O site of proline in the peptide GPG-NH_2_ was also evident from the UVRR data. This effect seemed to induce greater stabilization of the *trans* conformers of the peptide when it is dissolved in the KF and KCl water solutions with respect to the other solvation environments. The molecular picture offered by UVRR spectroscopy was consistent with the finding that the co-solute KF acts as a stabilizing agent on the β-turn structure of GPG-NH_2_, as detected by CD measurements. Overall, these results point out that the addition of different salts can favor the stabilization of the *trans* conformers in proline-based peptides. The application of the UVRR method to bioactive peptides containing X-Pro bonds could help in rationalizing the impact of salts as co-solutes on the *trans–cis* isomerization about the proline bond that is believed to be critical for the stability, dynamics and folding of proteins.

## Figures and Tables

**Figure 1 life-11-00824-f001:**
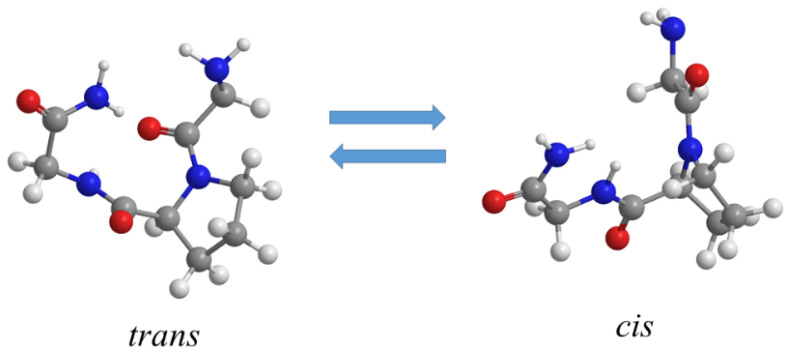
Molecular structure of the *trans* and *cis* conformers of GPG-NH_2_ tripeptide in an aqueous solution.

**Figure 2 life-11-00824-f002:**
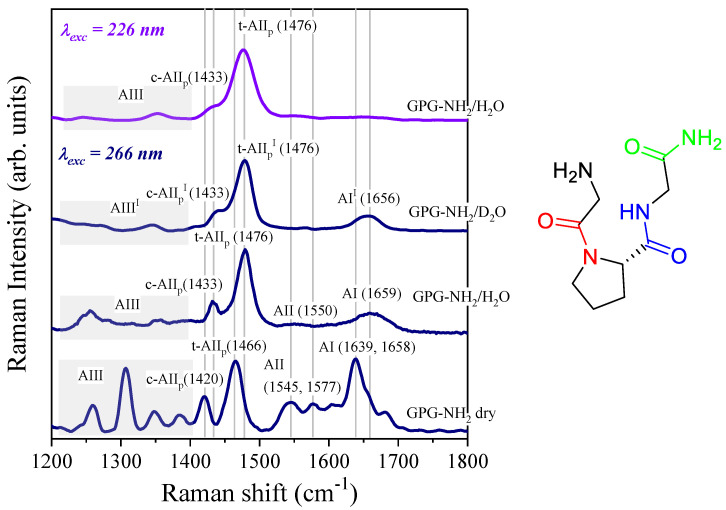
Bottom to top: 266 nm-excited spectra of lyophilized powder of GPG-NH_2_, GPG-NH_2_/H_2_O and GPG-NH_2_/D_2_O solutions and the 226 nm-excited spectrum of GPG-NH_2_/H_2_O recorded at 298 K. All the spectra have been normalized to their maximum value for a better comparison. Inset at the right: molecular structure of the GPG-NH_2_ tripeptide; the X-Pro peptide bond is highlighted in red with respect to the other amide groups (blue and green colored).

**Figure 3 life-11-00824-f003:**
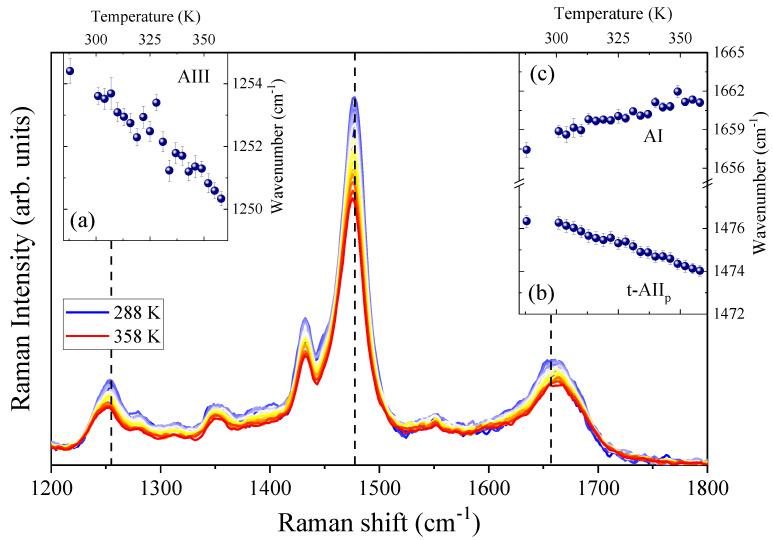
The 266 nm-excited spectra of hydrated GPG-NH_2_ (1:58 GPG-NH_2_:water) as a function of temperature in the range 288 K < T < 358 K. Insets: temperature dependence of the central wavenumber positions of the Raman bands of AIII (**a**), t-AII_p_ (**b**) and AI (**c**).

**Figure 4 life-11-00824-f004:**
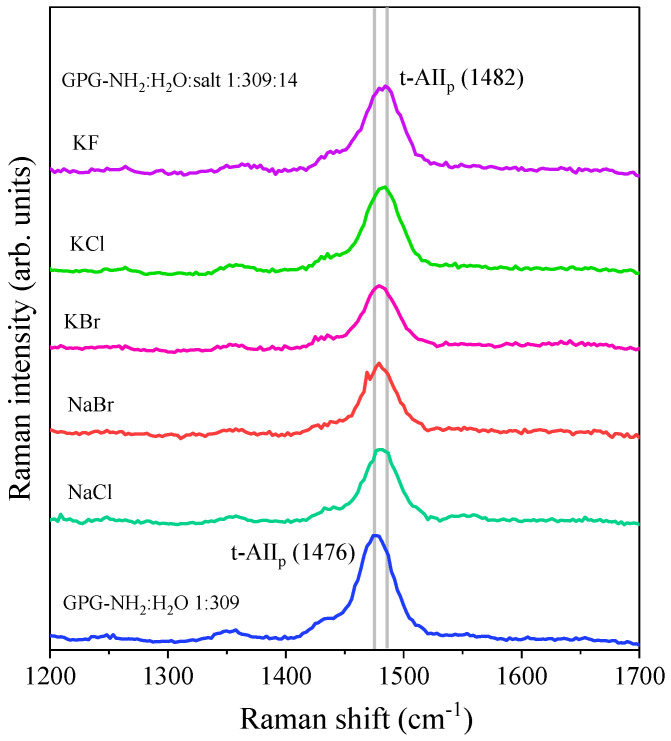
**The** 226 nm-excited spectra of hydrated GPG-NH_2_ (GPG-NH_2_:H_2_O 1:309) and ternary solutions GPG-NH_2_:H_2_O:salts (1:309:14) collected at T = 298 K.

**Figure 5 life-11-00824-f005:**
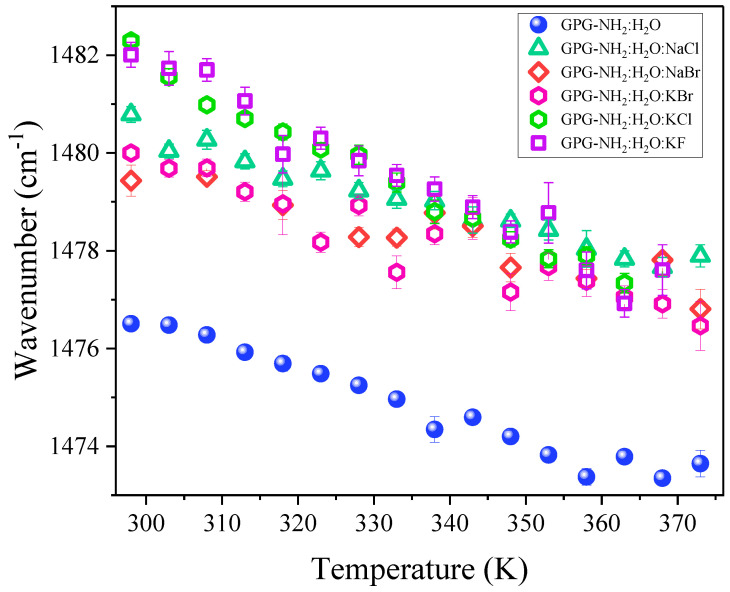
Temperature dependence of the wavenumber position of the t-AII_p_ band for hydrated GPG-NH_2_ (GPG-NH_2_:H_2_O 1:309) and ternary solutions of GPG-NH_2_:H_2_O:salts (1:309:14).

**Figure 6 life-11-00824-f006:**
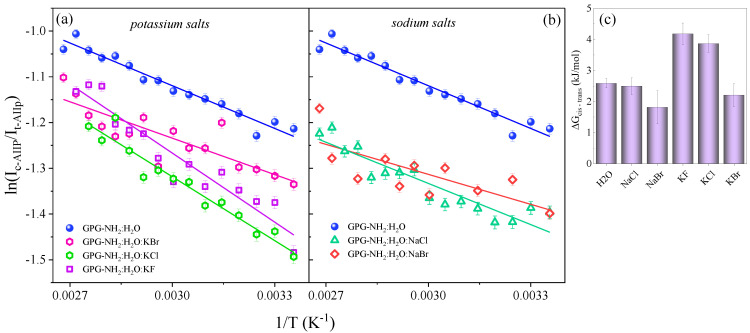
(**a**,**b**) Plot of the log of the *cis* to *trans* AmII_p_ intensity ratio versus 1/T for hydrated GPG-NH_2_ (GPG-NH_2_:H_2_O 1:309) and ternary solutions of GPG-NH_2_:H_2_O:salt (1:309:14); the solid lines represent the linear fit of the experimental data by following Eqn 1 (see text for details). (**c**) The free energy difference $\Delta G_{cis-trans}$ between the *cis* and *trans* conformations of GPG-NH_2_ estimated for the peptide dissolved in pure water and in water/salt solutions.

**Figure 7 life-11-00824-f007:**
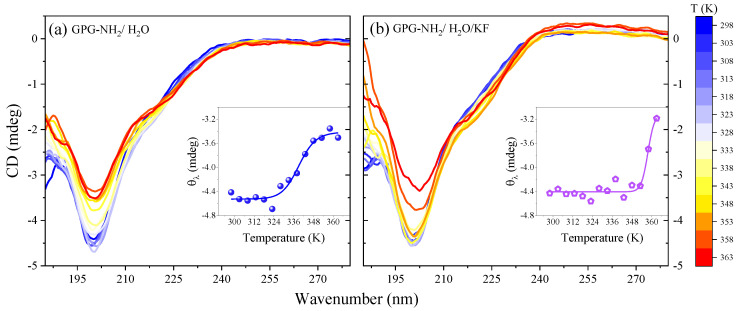
CD spectra as a function of temperature for GPG-NH_2_:H_2_O (**a**) and GPG-NH_2_:H_2_O:KF (**b**) solutions. Insets: temperature dependence of ellipticity θ_λ_ at 200 nm wavelength for the GPG-NH_2_:H_2_O and GPG-NH_2_:H_2_O:KF solutions.

**Table 1 life-11-00824-t001:** Wavenumber position of the t-AII_p_ band in hydrated GPG-NH_2_ in the presence of different salts at T = 298 K. The difference, Δν, is estimated with respect to the position of the t-AII_p_ band in the tripeptide dissolved in pure water ^1^.

Co-Solvent	Position of t-AII_p_ (cm^−1^)	Δν (cm^−1^)
NaCl	1480.8 ± 0.2	4.8
NaBr	1479.4 ± 0.3	3.4
KCl	1482.3 ± 0.1	6.3
KBr	1480.0 ± 0.1	4.0
KF	1482.0 ± 0.3	6.2

^1^ Values obtained for ternary solutions of GPG-NH_2_:H_2_O:salts (1:309:14) at T = 298 K.

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
