# Peer review of "Hydrogen Bonding and Solvation of a Proline-Based Peptide Model in Salt Solutions"

_life, 2021, doi:10.3390/life11080824_

Round 1
Reviewer 1 Report
In this manuscript, the authors revealed the effect of several salts on the hydration and intramolecular hydrogen bonding of a proline residue by selectively observing the proline amide bond in UV resonance Raman spectroscopy. It was suggested that KF and KCl significantly enhanced hydrogen bonding of the proline residue to stabilize the trans-isomer, which also contributed to stabilize a secondary structure of the tripeptide in aqueous solutions. These discussion seems to be reasonable based on several background studies shown in the references. Therefore, I think this manuscript is suitable for publishing in life, but the following minor points are better to be improved before the manuscript is accepted.
1. Lyophilized powder of the tripeptide was analyzed, but I could not understand that the powder was crystalline or amorphous, and hydrate or anhydrous forms. The presence or absence of hydrate waters may be important to assign the Raman signals of the lyophilized sample.
2. If there is a chemical structural formula of the tripeptide taking a b-turn conformation, I think readers can understand the contents more easily.
Author Response
Comments and Suggestions for Authors
In this manuscript, the authors revealed the effect of several salts on the hydration and intramolecular hydrogen bonding of a proline residue by selectively observing the proline amide bond in UV resonance Raman spectroscopy. It was suggested that KF and KCl significantly enhanced hydrogen bonding of the proline residue to stabilize the trans-isomer, which also contributed to stabilize a secondary structure of the tripeptide in aqueous solutions. These discussion seems to be reasonable based on several background studies shown in the references. Therefore, I think this manuscript is suitable for publishing in life, but the following minor points are better to be improved before the manuscript is accepted.
Lyophilized powder of the tripeptide was analyzed, but I could not understand that the powder was crystalline or amorphous, and hydrate or anhydrous forms. The presence or absence of hydrate waters may be important to assign the Raman signals of the lyophilized sample.
Author Response: we thank the reviewer for raising this point. As specified in the “Material and Methods” section of the revised manuscript: “…The tripeptide glycyl-L-prolyl-glycinamide HCl (GPG-NH2, MW=264.71 g/mol, > 99 % purity) is purchased from Bachem and used without further purification. No relevant contamination by water can be measured in the lyophilized powder of peptide as deducted by the absence in the Raman spectra of any signal attributable to the intense OH stretching band of water.”
If there is a chemical structural formula of the tripeptide taking a b-turn conformation, I think readers can understand the contents more easily.
Author Response: following the reviewer’s suggestion, we have added in the revised version of the manuscript a new figure (Figure 1) that shows the molecular structure of trans and cis GPG-NH2 molecule.
Reviewer 2 Report
In their manuscript, Rossi and coworkers describe the conformational analysis of the H-Gly-Pro-Gly-NH2 tripeptide in aqueous salt solutions by Raman spectroscopic. The authors show that depending on the anion and cation of the salt additive, differently strong intramolecular hydrogen bonds are formed, which influence the trans/cis ratio of the tertiary Pro amide bond.
The experimental data presented in the manuscript has been carefully analyzed and the suggested conclusions are of interest for the peptide chemistry community. Yet, I think that the conclusions drawn in the manuscript might be a bit too hasty and that additional experimental evidence is required. I would consider recommending publication of the manuscript after the following questions/points are addressed:
The authors attribute the signals at appox. 1430 cm-1 and 1475 cm-1 to the trans and cis configured Pro amide. Yet, I don’t see how this conclusion can be drawn from the data. Although the trans amide should be the major conformer, the extraction coefficients corresponding to the two transition could be different leading to unexpected signal intensities. I would therefore suggest NMR spectroscopic analysis of the peptide to confirm the presence of the two species (using nuclear Overhauser effect).
Furthermore, from the presented data, it is not clear which hydrogen bond is formed. Is it a type I or a type II beta turn. Also a gamma turn would be a possible option to form intramolecular hydrogen bonds. The authors should analyze the conformation of the peptide in solution by 1H NMR (extraction of the coupling constants) and nOe`s to identify the corresponding turn structure. Also a computational analysis of the peptide could be of help (also for the nOe analysis), however these often overestimate the stability of gamma-turns making spectroscopic evidence inevitable.
What is the role of the salt ions for the observed conformational changes? It is known that trans and cis conformer of Pro containing peptides differ in their dipole moments. Does the presence of the salt lead to a higher electric field around the peptide resulting in balancing the dipole moment of the peptide? Or is there a specific binding between peptide and ions? Also here, NMR spectroscopic and computational analyses could elucidate these questions.
In addition, I think the manuscript would benefit from an additional example, such as e.g. H-Gly-Pro-Ala-NH2 to show that the conclusions drawn in the manuscript are of a more general nature.
Author Response
Comments and Suggestions for Authors
In their manuscript, Rossi and coworkers describe the conformational analysis of the H-Gly-Pro-Gly-NH2 tripeptide in aqueous salt solutions by Raman spectroscopic. The authors show that depending on the anion and cation of the salt additive, differently strong intramolecular hydrogen bonds are formed, which influence the trans/cis ratio of the tertiary Pro amide bond.
The experimental data presented in the manuscript has been carefully analyzed and the suggested conclusions are of interest for the peptide chemistry community. Yet, I think that the conclusions drawn in the manuscript might be a bit too hasty and that additional experimental evidence is required. I would consider recommending publication of the manuscript after the following questions/points are addressed:
The authors attribute the signals at appox. 1430 cm-1 and 1475 cm-1 to the trans and cis configured Pro amide. Yet, I don’t see how this conclusion can be drawn from the data. Although the trans amide should be the major conformer, the extraction coefficients corresponding to the two transition could be different leading to unexpected signal intensities. I would therefore suggest NMR spectroscopic analysis of the peptide to confirm the presence of the two species (using nuclear Overhauser effect).
Author Response: Raman signals at ≈1430 cm-1 and 1475 cm-1 has been assigned to the trans and cis configured Pro amide of GPG-NH2 on the basis of the assignments reported in literature by extensive experimental and theoretical studies carried out on several peptides containing proline [see references 30 and 31 reported in the manuscript]. The presence of an 85:15 mixture the two species (trans and cis conformers) of GPG-NH2 in aqueous solution has been previously assessed by neutron diffraction enhanced by isotopic substitution (NDIS) in concert with NMR spectroscopy and both molecular dynamics (MD) and empirical potential structural refinement (EPSR) simulations and reported in the ref. 34 cited in the manuscript. Following the suggestion of the reviewer, this part of analysis has been improved in the revised version of the manuscript: “…The structure of GPG-NH2 in aqueous solution has been previously assessed using NMR spectroscopy and molecular dynamics (MD) simulations [34]. From those experiments it has been estimated an 85:15 mixture of trans:cis conformations (Fig. 1) about the Gly1-Pro2 peptide bond… Since GPG-NH2 dissolved in water contains both the cis and trans isomeric forms of the peptide [34,35], we associate the Raman signals observed at 1433 and 1476 cm-1 in the spectrum of the hydrated tripeptide with the vibrations of the proline AIIp band of the cis (c-AIIp) and the trans (t-AIIp) conformers, respectively [30,31]. This attribution is consistent with the assignments reported in previous works for cis and trans conformers of polyproline and other dipeptides containing proline [30,31]. Moreover, the downshift of ≈ 40 cm-1 of the proline AIIp band of the cis form of GPG-NH2 with respect to the trans one is in agreement with the observation that the amide II frequency of cis peptides is downshifted than that of trans peptides [39].”
Furthermore, from the presented data, it is not clear which hydrogen bond is formed. Is it a type I or a type II beta turn. Also a gamma turn would be a possible option to form intramolecular hydrogen bonds. The authors should analyze the conformation of the peptide in solution by 1H NMR (extraction of the coupling constants) and nOe`s to identify the corresponding turn structure. Also a computational analysis of the peptide could be of help (also for the nOe analysis), however these often overestimate the stability of gamma-turns making spectroscopic evidence inevitable.
Author Response: the formation and stabilization of turns in the structure of GPG-NH2 peptide have been extensively investigated by NMR and computational methods and the results are reported in the paper of Busch et al. (ref. 34 cited in the manuscript). Moreover, recent neutron scattering experiments have been used for providing the distribution of the intra- and inter-molecular hydrogen bonds formed by GPG-NH2 that are involved in the formation of the b-turn (ref 35 cited in the manuscript). The aim of the present work is to investigate the hydrogen bond interaction between water molecules and proline residue of GPG-NH2 that is affected but not directly involved in the formation of turn in the peptide.
What is the role of the salt ions for the observed conformational changes? It is known that trans and cis conformer of Pro containing peptides differ in their dipole moments. Does the presence of the salt lead to a higher electric field around the peptide resulting in balancing the dipole moment of the peptide? Or is there a specific binding between peptide and ions? Also here, NMR spectroscopic and computational analyses could elucidate these questions.
Author Response: the point raised by the reviewer has been already addressed in the work of Busch et al. “Busch, S., Pardo, L.C., O'Dell, W.B., Bruce, C.D., Lorenz, C.D. and McLain, S.E., 2013. On the structure of water and chloride ion interactions with a peptide backbone in solution. Physical Chemistry Chemical Physics, 15(48), pp.21023-21033”. In the latter paper, Molecular Dynamics (MD) simulations and neutron diffraction experiments are used for investigate the arrangement of water and chloride ions around the GPG-NH2 peptide. The results evidence two different types of chloride interactions, namely a direct contact between Cl- and the peptide backbone and a water-mediated interaction, that seem to common for amine and amide groups.
The aim of our work is to provide a molecular characterization of the GPG-NH2 peptide structure and of the variations of the hydration shell in the presence of different salts using UV Resonance Raman technique (see sections “5.2 Effect of salts on the hydrogen bonding at the carbonyl site of Pro” and “5.3 Influence of salts on trans/cis isomerization” in the manuscript).
In addition, I think the manuscript would benefit from an additional example, such as e.g. H-Gly-Pro-Ala-NH2 to show that the conclusions drawn in the manuscript are of a more general nature.
Author Response: we agree with the reviewer that the UVRR experimental approach highlighted in the manuscript could be applied to the investigation of other proline-based peptides in order to rationalize the effect of ion-peptide interactions in affecting the conformational stability and the trans-cis equilibrium of these molecules. However, as specified in the manuscript “…the tripeptide glycyl-L-prolyl-glycinamide·HCl (GPG-NH2) has been chosen for this study as a very simplified model system” and the investigation of other peptides, such as H-Gly-Pro-Ala-NH2, will be certainly the subject of next experiments.
Round 2
Reviewer 2 Report
I thank the authors for carefully addressing my concerns. Indeed, when the raised points were investigated in the original literature by Busch et al, then this is satisfying for the manuscript. The additions made improve the manuscript and make it a nice read!
Yet, considering how much has been done on the peptide (by Busch et al.) I am wondering what's the essential new message of the manuscript presented? Is it the sheer presentation of a new technique to analyze Pro-containing peptides? This would suggest a more analytics focus journal for the manuscript. If the authors can provide a conclusive statement about novel insights that bring the community a step forward in the understanding of peptide solvation, I am happy to suggest acceptance.
Author Response
Comments and Suggestions for Authors
I thank the authors for carefully addressing my concerns. Indeed, when the raised points were investigated in the original literature by Busch et al, then this is satisfying for the manuscript. The additions made improve the manuscript and make it a nice read!
Yet, considering how much has been done on the peptide (by Busch et al.) I am wondering what's the essential new message of the manuscript presented? Is it the sheer presentation of a new technique to analyze Pro-containing peptides? This would suggest a more analytics focus journal for the manuscript. If the authors can provide a conclusive statement about novel insights that bring the community a step forward in the understanding of peptide solvation, I am happy to suggest acceptance.
Author Response: we thank the reviewer for the appreciation of the improvements made in the revised manuscript. We agree with the reviewer that an extended characterization of the conformation and hydration shell of GPG-NH2 peptide in water has been reported by Busch at al. However, as highlighted in the abstract and in the conclusion of the manuscript, the focus of the present work is to provide insights on the role played by ion-peptide interactions in affecting the trans-cis equilibrium in a proline-based peptide model.
Following the suggestion of the reviewer, some sentences have been added in the conclusion of the revised manuscript that highlight the novel insights of the present work: “Overall these results point out that the addition of different salts can favour the stabilization of the trans conformers in proline-based peptides. The application of the UVRR method to bioactive peptides containing X-Pro bonds could help to rationalize the impact of salts as co-solutes on the trans-cis isomerization about the proline bond that is believed to be critical for the stability, dynamics and folding of proteins.”